# Comprehensive Genetic Analysis of Druze Provides Insights into Carrier Screening

**DOI:** 10.3390/genes14040937

**Published:** 2023-04-18

**Authors:** Eden Avnat, Guy Shapira, Shelly Shoval, Ifat Israel-Elgali, Anna Alkelai, Alan R. Shuldiner, Claudia Gonzaga-Jauregui, Jamal Zidan, Taiseer Maray, Noam Shomron, Eitan Friedman

**Affiliations:** 1Faculty of Medicine, Tel Aviv University, Tel Aviv 69978, Israel; 2Edmond J. Safra Center for Bioinformatics, Tel Aviv University, Tel Aviv 69978, Israel; 3The Susanne Levy Gertner Oncogenetics Unit, Institute of Human Genetics, Sheba Medical Center, Ramat Gan 52621, Israel; 4Sagol School of Neuroscience, Tel-Aviv University, Tel Aviv 69978, Israel; 5Regeneron Genetics Center, Tarrytown, NY 10591, USA; 6International Laboratory for Human Genome Research, Laboratorio Internacional de Investigación sobre el Genoma Humano, Universidad Nacional Autónoma de México, Juriquilla 04510, Querétaro, Mexico; 7The Oncology Department, Ziv Medical Center, and the Azrieli Faculty of Medicine, Bar-Ilan University, Zefat 13206, Israel; 8Golan for Development, Majdal Shams 1243800, Golan Heights; 9The Meirav High Risk Clinic, Chaim Sheba Medical Center, Tel-Hashomer, Ramat Gan 52621, Israel

**Keywords:** druze, genetic isolate, founder population, whole exome sequencing, recurring pathogenic variants

## Abstract

Background: Druze individuals, like many genetically homogeneous and isolated populations, harbor recurring pathogenic variants (PV) in autosomal recessive (AR) disorders. Methods: Variant calling of whole-genome sequencing (WGS) of 40 Druze from the Human Genome Diversity Project (HGDP) was performed (HGDP-cohort). Additionally, we performed whole exome sequencing (WES) of 118 Druze individuals: 38 trios and 2 couples, representing geographically distinct clans (WES-cohort). Rates of validated PV were compared with rates in worldwide and Middle Eastern populations, from the gnomAD and dbSNP datasets. Results: Overall, 34 PVs were identified: 30 PVs in genes underlying AR disorders, 3 additional PVs were associated with autosomal dominant (AD) disorders, and 1 PV with X-linked-dominant inherited disorder in the WES cohort. Conclusions: The newly identified PVs associated with AR conditions should be considered for incorporation into prenatal-screening options offered to Druze individuals after an extension and validation of the results in a larger study.

## 1. Introduction

Druze individuals constitute a Middle Eastern minority population. Traditionally, the Druze religion is believed to have formed as an Islamic reform movement, under the rule of the sixth caliph of the Fatimid Dynasty of Egypt, ElHakim (AD 966–1020) [1]. In Israel, there are ~150,000 Druze (of an estimated ~1,000,000 worldwide), overwhelmingly residing in the Northern part of the country [2]. For centuries, Druze have strictly prohibited marriage to non-Druze and limited conversion into the religion. These practices, combined with a high rate of (47%) consanguineous marriages [3], and residence in isolated, mountainous regions, have made the Druze a unique population for genetic research.

Given the founder population attributes of Druze, drifted variants resulting in a high prevalence of monogenic disorders are expected. Indeed, previously reported recurring pathogenic variants (PVs) amongst Druze include two PVs in the *ATM* gene (the gene that underlies Ataxia Telangectesia–OMIM # 208900) in Druze communities in Jordan, Lebanon, and Syria [4]; a PV in the β globin gene [5]; and a nonsense variant in the LDL receptor (LDLR) gene, causing familial hypercholesterolemia [6]. In the most comprehensive account of prevalent germline PVs causing autosomal recessive (AR) disorders in the non-Jewish Israeli population, of 103 PVs in 81 genes, 32 PVs were founder mutations in Druze individuals [7].

Behar et al. [8] demonstrated close genetic relations between Druze and other Middle Eastern populations, such as Bedouins, Palestinians, Syrians, Lebanese, and Jews. A previous study published by some of us [9] confirmed the Middle Eastern origins of the Druze, as well as suggested a ≈ 15-fold reduction in population size taking place ≈ 22–47 generations ago.

In the current study, we performed whole exome sequencing (WES) in 118 samples collected from Druze trios SNP-genotyped in our previous study [9] to further define the genetic makeup of Druze individuals and characterize novel, clinically relevant coding variants in this population. We also analyzed HGDP-available Druze whole-genome sequence (WGS) data from 40 distinct Druze samples [10] (Figure 1).

## 2. Materials and Methods

### Recruitment of Druze Participants for WES

Druze trios—The study population was individuals who were recruited and participated in our previously described study [9]. Briefly, in the original study, 40 trios of Druze origin (*n* = 120) representing the different clans (Hamullas) were recruited. These healthy participants were recruited from the Druze communities in Beit Jan located in the Northern Galilee in Israel (20 trios) and in the Golan Heights (20 trios), primarily the village of Majdal Shams. Clan ancestral roots were based on family names and repress ented the origins of major locales of Druze residing in the Middle East. Only 118/120 individuals recruited in the original study were included herein, based on DNA quality and availability. HGDP cohort—The HGDP contains 929 DNA samples and WGS data from ethnically diverse individuals, including 40 Druze samples [10]. HGDP DNA samples were Illumina-genome sequenced to an average coverage of 35× (minimum 25×) and reads were mapped to the GRCh38 reference assembly as reported [10]. HGDP Druze study individuals resided in Druze villages in the Carmel and Galilee regions of Israel and not in the Golan Heights. Whole exome sequencing—WES was carried out at the Regeneron Genetics Center following previously published protocols [11]. In brief, genomic DNA was sheared and used to prepare 75 bp paired-end libraries for exome sequencing. Samples were captured using the IDT XGen exome capture reagent and sequenced on an Illumina NovaSeq instrument. Captured fragments were sequenced to achieve a minimum of 85% of the target bases covered at 20× or greater. Following sequencing, data were processed using a DNAnexus-implemented cloud-based pipeline that runs standard tools for sample-level data production and analysis. Sequence reads were mapped and aligned to the GRCh38/hg38 human genome reference assembly using BWA-mem and SNP and InDel variants, and genotypes were called using GATK’s HaplotypeCaller in accordance with the best practices for germline short-variant discovery. Samtools 1.12 was used for coverage and depth calculations. Variant filtering—In this study we focused on variants that were labeled as either “Pathogenic” or “Likely pathogenic” (PV) according to ClinVar (https://www.ncbi.nlm.nih.gov/clinvar/ (accessed on 23 January 2023)). Additionally, the actual pathogenicity of each PV was classified according to the American College of Medical Genetics and Genomics and the Association for Molecular Pathology (ACMG-AMP) guidelines [12]. Since the focus of this research is on disease-associated variants previously unreported in the Druze population, variants previously reported to be present in the Druze population and appear in previous relevant studies or at the Israeli Medical Genetic Database (http://INGD.huji.ac.il) are listed separately in Appendix A (WES analysis) and Appendix A (HGDP analysis).

Sources of comparison populations and datasets—For each PV, general population allele count (AC) and general population allele number (AN) were retrieved from gnomAD (https://gnomad.broadinstitute.org/ (accessed on 1 November 2022)), as indicated by the total row in the Population Frequencies table. If AC and AN were missing, those values were extracted from dbSNP (https://www.ncbi.nlm.nih.gov/snp/ (accessed on 1 November 2022)), as indicated by the total column in the ALFA allele-frequency table. Additionally, suitable AC and AN of the Middle Eastern population were extracted from gnomAD, as indicated by the Middle East row in the population frequencies table. Allele frequency (AF) was calculated by dividing AC by AN.

Statistical analyses—A two-sided Fisher’s exact test was performed to compare the difference between the AF of the WES cohort and the AF of the general population for each SNP and between the AF of the HGDP cohort and the AF of the general population for each SNP. A *p* value of 0.05 was set to be the cutoff for statistically significant results.

## 3. Results

### 3.1. WES Analysis

Overall, 118 Druze individuals had WES analysis: 46% were females (54/118), 44% were males (52/188) and 10% were missing sex data. The age range for this cohort was 14–60 years (median 34 years). Overall, 20 PVs were detected in the WES analysis (18 AR PVs, 1 AD PV, and 1 XLD PV): 40% (8/20) were missense variants, 25% (5/20) were frameshift variants, 20% (4/20) were nonsense (stop gain) variants, 10% (2/20) were in-frame deletion variants, and 5% (1/20) was a splice-acceptor variant (Table 1) (Figure 2). Fisher’s exact test was not performed on rs1555400381 since no relevant general population AF was found.

### 3.2. HGDP Analysis

Forty Druze individuals had WGS: 73% females (29/40) and 27% males (11/40). Overall, 15 PVs were detected in the HGDP analysis (13 AR PV and 2 AD PV); 40% (6/15) were nonsense (stop gain) variants, 27% (4/15) were missense variants, 27% (4/15) were frameshift variants, and 6% (1/15) were splice donor variants (Table 2, Figure 3).

One PV (rs777172978) was significantly enriched in both the WES and the HGDP analysis.

In the HGDP cohort, we focused on three noncoding SNPs spanning 9118 bps located in the IL18R1 gene (OMIM #604494): rs12987977 (nucleotide alteration NM_003855.5: c.-29 + 2476T > G); rs12999364 (nucleotide alteration NM_003855.5: c.-29 + 1269C > T); and rs4851569 (nucleotide alteration NM_003855.5: c.59-1038C > A). In Druze, these three SNPs were in linkage disequilibrium with each other: rs12987977 and rs4851569 (D’ = 0.99, R2 = 0.90 and *p*-value < 0.0001), rs12987977 and rs12999364 (D’ = 1.0 R2 = 0.998 and *p*-value < 0.0001) and rs12999364 and rs4851569 (D’ = 1.0 R2 = 0.90 and *p*-value < 0.0001). All three SNPs (rs12987977, rs12999364, rs4851569) were significantly more prevalent in the Druze population (AF = 0.49, 0.49, and 0.49, respectively) compared with other populations (AF = 0.32, 0.32, and 0.34 respectively) (*p*-values = 0.002, 0.002 and 0.006, respectively). All three SNPs shared the same zygosity in all genotyped samples. In 10/40 samples all three SNPs were homozygous, and in an additional 21/40 samples, they were heterozygous. Notably, no protein truncating or pathogenic variants in this gene were noted in any individuals in the WES cohort.

## 4. Discussion

In the current study, 34 PVs in genes associated with AR and AD disorders not previously described in Druze individuals were identified. The most updated list of genes and PVs prevalent in the Druze population in Israel encompasses 79 AR diseases, 81 genes, and 103 variants [7]. The findings of PVs in the isolated populations reported herein are in line with previous reports [12,13]. Specifically, Khayat et al. [13] reported 48 PVs in the AR genes (24 novel PVs) in an isolated community of Muslim Arabs in Israel (*n* = 50) based on the results of WES in that population [14]. The Israeli population genetic carrier screening program is included in the health basket and hence is covered by the health maintenance organizations (HMOs) [15]. The data presented herein suggest that the expansion of the list of testable AR disease genes genotyped in the context of the Israeli population genetic-carrier screening program should be considered. Such a list should be based on more comprehensive data collected from all ethnicities with a specific emphasis on genotyping adequate numbers of individuals from isolated populations to address their unique needs. Notably, rates of carrier screening use among Druze and other non-Jewish ethnic groups in Israel are substantially lower compared to rates in Jewish Israeli counterparts [16]. Given the cost effectiveness of prenatal screens in guiding prenatal diagnostic procedures, awareness of the availability of effective testing should be increased in the Druze population.

PVs in two genes that are associated with AD chronic pancreatitis-*PRSS1* (OMIM #276000; PV-rs111033565) and *CTRC* (OMIM #601405; PV-rs202058123) were detected. The incidence of chronic pancreatitis ranges from 4 to 14 per 100,000 per year, and the prevalence from 13 to 52 per 100,000 population [17]. There are no reported studies suggesting that Druze individuals are at an increased risk for developing chronic pancreatitis compared with other ethnically diverse populations. Since clinical manifestations may be subtle, the implication of this finding needs to be investigated in a larger population of Druze cases. Perhaps those that are referred for a clinical workup of undefined abdominal pain or nonspecific symptom that may herald chronic pancreatitis. Other possibilities to account for these genetic findings, as well as for other seemingly prevalent PVs in AD disorders reported herein, should also be entertained: incomplete penetrance, or even misclassification of pathogenicity by ClinVar.

Notably, the high rate of the PV in the *PRRT2* gene in the current study (3%), as is the rate of the PV in the *COL6A2* gene (3%), are expected to be associated with a high rate of Episodic Kinesigenic Dyskinesia, Type 1 and Ullrich congenital muscular dystrophy, Type 1 amongst Druze individuals, respectively. Underreporting, incomplete penetrance, or variable expressivity of these disorders in Druze individuals, as indeed is the case in other populations for Episodic Kinesigenic Dyskinesia, Type 1 [18], may account for the lack of reported overrepresentation of clinically relevant diseases.

In this study, we identified two PVs in two ACMG actionable genes [19] *MUTYH* (OMIM #604933; PV-rs587778541) and *MEFV* (OMIM #608107; PV-rs28940580). Homozygous *MUTYH* PVs are associated with colorectal cancer and adenomatous polyposis while homozygous PVs in MEFV cause Familial Mediterranean Fever (FMF), a relatively prevalent disease in people who live around the Mediterranean region, including the Druze population [20,21]. Notably, homozygous PVs in both genes are associated with a clinically significant disease, whereas heterozygous PVs, as is the case here, are not.

The p.I1307K *APC* (OMIM #611731) increased risk allele was detected in two Druze, cancer-free individuals in the current study (AF = 0.03, AC = 2). This variant is very prevalent in Ashkenazi Jews (AJ), ~6% [22] of the general average risk population with rates of up to 20% in AJ colorectal cancer (CRC) cases with a family history of CRC [23]. Since its original description in AJ, this variant has been reported in ethnically diverse populations of Jewish non-Ashkenazim [24] and Muslim Arabs residing in Israel [25]. Detecting this variant in Druze individuals, given the unique and almost exclusive intrafaith marriage patterns, may suggest that this variant may have arisen in the Middle East prior to the separation of the Druze from the Muslims. The clinical implication of harboring the p.I1307K *APC* variant and the associated cancer risk is still unsettled. In most studies, this variant marginally increases the risk for developing CRC with a pooled odds ratio in one meta-analysis of 2.17 (95% confidence interval: 1.64, 2.86) [26] with the median age not younger in variant carriers compared to the general population [27]. The risk for developing CRC in Israel is significantly lower for non-Jewish individuals compared with ethnically diverse Jews (https://www.health.gov.il/UnitsOffice/HD/ICD C/ICR/CancerIncidence/Pages/default.aspx (accessed on 1 November 2022)). Yet the carrier rate in non-AJ of the p.I1307K *APC* variant is estimated to be 1.6% [27], similar to what has been observed in the current study. Taken together, these facts may be indirect evidence for a minimal role of this specific *APC* variant in conferring CRC risk during population screens.

Behçet disease (BD) is a multisystem inflammatory disorder pathologically hallmarked by vasculitis affecting the small and large veins and arteries [28]. Ethnic groups living along the historical silk road are at an increased risk of developing BD [29]. Specifically, in Israel, the rate of BD amongst Druze is reportedly among the highest of all ethnic groups with rates of up to 150/100,000 [8]. Like most adult-onset diseases, genetic factors play a role in BD predisposition. Notably, human leukocyte antigen (HLA)-B51 has been reported as the strongest genetically-associated factor for BD. Other HLA alleles, as well as other loci containing genes involved in host defense, immunity, and inflammation pathways (detected predominantly via GWAS), have been shown to contribute to BD susceptibility [30]. Of these additional BD-associated genes, the interleukin pathway family of genes, including *IL10*, *IL23R-IL12RB2*, *IL12A*, and *IL23R*, have been reported [30]. Specifically, the possible contribution of the *IL18R1* gene to BD has not been thoroughly investigated. *IL18R1* encodes for the α chain, a subunit of the IL18 receptor [31]. IL18, the IL18 receptor ligand, is a member of the IL1 family of cytokines [31], proteins that play a key role in BD ocular or mucocutaneous manifestations and was found to be elevated in the synovial fluid of BD patients [32,33]. Tan and coworkers [34] reported that three SNPs in the genomic region encompassing the *IL18R1* gene were associated with ocular manifestations of BD in the Han Chinese population. In the current study, these three SNPs were in perfect linkage disequilibrium creating a 10 Kb haplotype enriched in the Druze population. Yet, the high rate of these SNPs in the general population, the lack of any bona fide PVs in the WES cohort, and the paucity of supporting data in other populations may indicate that the contribution of PVs in the *IL18R1* gene to the burden of BD may be minimal at best.

The limitations of the current study should be acknowledged. This is a study that generated data on a limited number of Druze families residing in Israel, where only a small subset of the world Druze population resides, and it may not reflect the entire populational spectrum of this ethnic community. Given the lack of precise clinical knowledge on the genotyped individuals and basing the health status on self reporting at a single time point adds another limitation. Given the current study design, the penetrance of the autosomal dominant alleles reported herein cannot be assessed, thus limiting the ability to provide more insightful and evidence-based genetic counseling. Additionally, the results on which AR genes’ PVs (or a subset of them) should be incorporated into a Druze prenatal screening, should await a validation study encompassing more Druze cases.

## 5. Conclusions

Novel PVs in genes associated with severe AR disorders prevalent in Druze individuals should be considered for inclusion in the next version of the national prenatal screening in Israel to the relevant population, after validation in a larger study.

## Figures and Tables

**Figure 1 genes-14-00937-f001:**
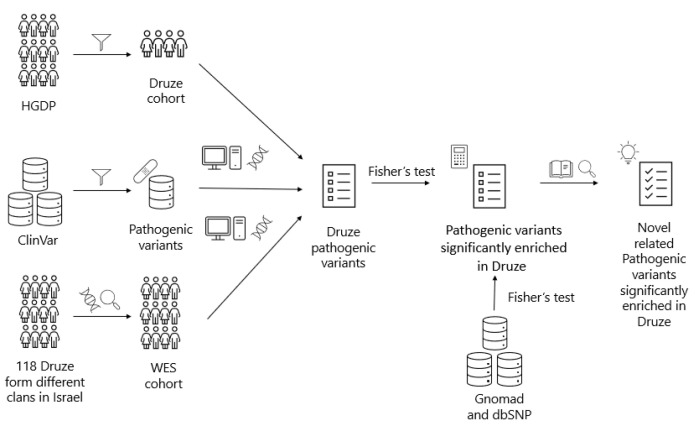
Methodology flow diagram: HGDP-derived data was filtered based on Druze ethnicity to create a Druze cohort of 40 individuals. Additionally, exome sequencing was performed on 118 Druze individuals from different clans in Israel, creating the WES cohort. Simultaneously, all the variants from ClinVar were filtered based on interpretation labeled as “pathogenic” or “likely pathogenic”. Then, the Druze-cohort variants and the WES-cohort variants were cross referenced with the catalogue of the pathogenic variants from ClinVar creating the Druze pathogenic-variants list. Only variants that were classified as “pathogenic” or “likely pathogenic” according to the ACMG-AMP guidelines were included in the list. We compared the allele frequency of each variant in our cohort and the allele frequency of the variants in worldwide populations based on the data from gnomAD and dbSNP. Using Fisher’s test, we identified the variants that were significantly different in Druze. After a literature review, we narrowed down the list to obtain a curated set of pathogenic variants that are enriched in the Druze population in comparison to other populations.

**Figure 2 genes-14-00937-f002:**
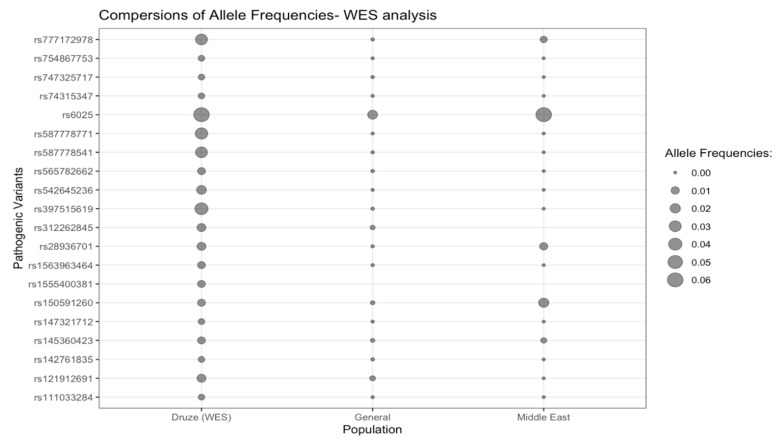
Bubble plot representing the AF of each PV according to the WES analysis and with respect to Druze, General, and Middle East populations. Missing allele frequencies are represented by missing bubbles.

**Figure 3 genes-14-00937-f003:**
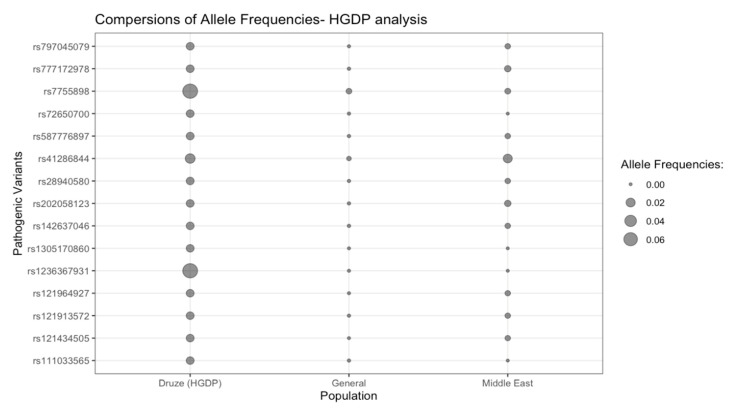
Bubble plot representing the AF of each PV according to the HGDP analysis and with respect to Druze, General, and Middle East populations. Missing allele frequencies are represented by missing bubbles.

**Table 1 genes-14-00937-t001:** Potentially clinically relevant genes and variants in Druze from exome sequencing analysis.

Rs Number	ClinVar ID ^a^	Condition ^b^	Gene (OMIM #)	Nucleotide Alteration (Amino Acid)	Mutation Type	Location ^c^	Durze AF (AC/AN)	Heterozygous/Homozygous	General Population AF (AC/AN)	Fisher’s *p*-Value	Middle East AF (AC/AN)
rs142761835	167199 (2, P)	Isovaleryl-CoA dehydrogenase deficiency (AR)	IVD (607036)	NM_002225.5:c.358G > A (NP_002216.3:p.Gly120Arg)	Missense	15-40410699 G > A	0.004 (1/236)	1/0	0.0001 (17/152,182)	0.03	0 (0/316)
rs747325717	916358 (1, LP)	Charcot-Marie-Tooth disease axonal, type 2F (AD)	HSPB1 (602195)	NM_001540.5:c.438dup (NP_001531.1:p.Gly147fs)	Frameshift	7-76303987 G > GC	0.004 (1/236)	1/0	7 × 10^−5^ (11/152,154)	0.02	0 (0/316)
rs147321712	65804 (2, P)	OTOF-Related Deafness (AR)	OTOF (603681)	NM_194248.3:c.4483C > T (NP_919224.1:p.Arg1495Ter)	Nonsense (stop gain)	2-26466731 G > A	0.004 (1/236)	1/0	2 × 10^−5^ (3/152,210)	0.006	0 (0/316)
rs74315347	5368 (2, P)	Nephrotic syndrome, type 2 (AR)	NPHS2 (604766)	NC_000001.11:g.179557227C > T (NP_055440.1:p.Val180Met)	Missense	1-179557227 C > T	0.004 (1/236)	1/0	1 × 10^−5^ (2/152,126)	0.005	0 (0/316)
rs111033284	43340 (2, P)	Inborn genetic diseases (AR)	MYO7A (276903)	NM_000260.4:c.722G > A (NP_000251.3:p.Arg241His)	Missense	11-77156991 G > A	0.004 (1/236)	1/0	1 × 10^−5^ (2/152,244)	0.005	0 (0/316)
rs754867753	216118 (2, P)	Primary ciliary dyskinesia (AR)	CCDC40 (613799)	NM_017950.4:c.961C > T (NP_060420.2:p.Arg321Ter)	Nonsense (stop gain)	17-80050085 C > T	0.004 (1/236)	1/0	7 × 10^−6^ (1/151,790)	0.003	0 (0/316)
rs1555400381	558720 (2, P)	Propionic acidemia (AR)	PCCA (232000)	NM_000282.4:c.843del (NP_000273.2:p.Asn281fs)	Inframe deletion	13-100920966 AT > A	0.008 (2/236)	2/0	NA	NA	NA
rs150591260	203805 (2, P)	3-methylcrotonyl CoA carboxylase 2 deficiency (AR)	MCCC2 (609014)	NM_022132.5: c.1015G > A (NP_071415.1:p.Val339Met)	Missense	5-71641018 G > A	0.008 (2/236)	2/0	0.0006 (95/152,168)	0.01	0.02 (6/312)
rs145360423	426990 (2, P)	Chronic granulomatous disease, cytochrome b-positive, type 1 (AR)	NCF1 (608512)	NM_000265.7:c.579G > A (NP_000256.4:p.Trp193Ter)	Nonsense (stop gain)	7-74783529 G > A	0.008 (2/236)	2/0	0.0006 (84/151,994)	0.008	0.003 (1/316)
rs565782662 ^d^	5268 (2, P)	Netherton syndrome (AR)	SPINK5 (605010)	NM_006846.3:c.2468dupA (NP_006837.2:p.Lys824fs)	Frameshift	5-148120311 G > GA	0.008 (2/236)	2/0	0.0001 (15/146,014)	0.0003	0 (0/312)
rs1563963464 ^d^	634641 (1, P)	Ataxia-oculomotor apraxia type 1 (AR)	APTX (606350)	NC_000009.12:g.32986031C > A	Splice acceptor	9-32986031 C > A	0.008 (2/236)	2/0	0.00004 (3/82,310)	8 × 10^−5^	0 (0/118)
rs121912691	18115 (2, P)	Cystinuria (AR)	SLC3A1 (104614)	NM_000341.4:c.1400T > C (NP_000332.2:p.Met467Thr)	Missense	2-44312653 T > C	0.01 (3/236)	3/0	0.002 (358/152,198	0.02	0 (0/316)
rs312262845 ^d^	41142 (1, P)	Orofaciodigital syndrome I (XLD)	OFD1 (311200)	NM_003611.3:c.710del (NP_003602.1:p.Lys237fs)	Frameshift	X-13746826 CA > C	0.01 (3/236)	3/0	0.001 (111/85,096)	0.004	NA
rs28936701	7733 (2, P)	Glaucoma 3A (AR)	CYP1B1 (601771)	NC_000002.12:g.38070949G > A	Missense	2-38070949 G > A	0.01 (3/236)	3/0	5.3 × 10^−5^ (8/152,118)	6 × 10^−7^	0.01 (3/316)
rs542645236	225134 (3, P)	Phenylketonuria (AR)	PAH (612349)	NM_000277.3:c.320A > G (NP_000268.1:p.His107Arg)	Missense	12-102894767 T > C	0.02 (4/236)	4/0	1 × 10^−5^ (2/152,204)	8.4 × 10^−11^	0 (0/316)
rs777172978	1029383 (1, P)	Ullrich congenital muscular dystrophy, type 1 (AR)	COL6A2 (120240)	NM_058174.3:c.2554C > T (NP_478054.2:p.Gln852Ter)	Nonsense (stop gain)	21-46129288 C > T	0.03 (7/236)	7/0	7.2 × 10^−5^ (11/152,254)	6.10 × 10^−16^	0.006 (2/316)
rs587778541	127838 (2, P)	Hereditary cancer-predisposing syndrome (AR)	MUTYH (604933)	NM_001048174.2:c.1350GGA [1] (NP_001041639.1:p.Glu452del)	Inframe deletion	1-45331218 TTCC > T	0.03 (7/236)	7/0	5 × 10^−5^ (8/152,112)	1.2 × 10^−16^	0 (0/316)
rs587778771 ^d^	39752 (2, LP)	Episodic kinesigenic dyskinesia, type 1 (AR)	PRRT2 (614386)	NM_145239.3:c.649del (NP_660282.2:p.Arg217fs)	Frameshift	16-29813694 GC > G	0.03 (8/236)	8/0	1 × 10^−5^ (2/149,480)	1.5 × 10^−21^	0 (0/310)
rs397515619	6306 (1, P)	Spermatogenic failure, type 5 (AR)	AURKC (603495)	NM_001015878.2: c.145del (NP_001015878.1:p.Leu49fs)	Frameshift	19-57232069 TC > T	0.04 (9/236)	9/0	0.0001 (14/151,932)	3.6 × 10^−20^	0 (0/316)
rs6025	642 (2, P)	Thrombophilia due to factor V Leiden (AD)	F5 (612309)	NM_000130.5:c.1601G > A (NP_000121.2:p.Arg534Gln)	Missense	1-169549811 C > T	0.06 (14/236)	14/0	0.02 (2638/152,200)	8 × 10^−5^	0.06 (19/316)

^a^ Number of stars by ClinVar and pathogenic level according to the ACMG-AMP guidelines (P-Pathogenic or LP-Likely Pathogenic). ^b^ Autosomal Dominant = AD; Autosomal Recessive = AR; X-linked Dominant = XLD; Multifactorial = M. ^c^ Chromosome-Position Reference > Alternative. ^d^ Variant in a low complexity region according to GnomAD. NA-missing AF.

**Table 2 genes-14-00937-t002:** Potentially clinically relevant genes and variants from HGDP genome sequencing analysis.

Rs Number	ClinVar ID ^a^	Condition ^b^	Gene (OMIM #)	Nucleotide Alteration (Amino Acid)	Mutation Type	Location ^c^	Durze AF (AC/AN)	Heterozygous/Homozygous	General Population AF (AC/AN)	Fisher’s *p*-Value	Middle East AF (AC/AN)
rs1305170860	95704 (2, P)	Cobalamin C disease (AR)	MMACHC (609831)	NM_015506.3:c.547_548del (NP_056321.2:p.Val183fs)	Frameshift	1-45508909 CTG > C	0.01 (1/80)	1/0	6.6 × 10^−6^ (1/152,210)	0.001	0 (0/316)
rs121964927	265135 (2, P)	Congenital factor VII deficiency (AR)	F7 (613878)	NM_019616.4:c.1043G > T (NP_062562.1:p.Cys348Phe)	Missense	13-113118716 G > T	0.01 (1/80)	1/0	1.3 × 10^−5^ (2/152,168)	0.001	0.003 (1/316)
rs72650700	30339 (2, P)	Pseudoxanthoma Elasticum (AR)	ABCC6 (603234)	NM_001171.6:c.1552C > T (NP_001162.5:p.Arg518Ter)	Nonsense (stop gain)	16-16190247 G > A	0.01 (1/80)	1/0	4.6 × 10^−5^ (7/152,070)	0.004	0 (0/316)
rs28940580	36507 (2, P)	Familial Mediterranean Fever (AR)	MEFV (608107)	NM_000243.3:c.2040G > C (NP_000234.1: p.Met680Ile)	Missense	16-3243447 G > C	0.01 (1/80)	1/0	4.6 × 10^−5^ (7/152,148)	0.004	0.003 (1/316)
rs587776897	31015 (2, P)	Desbuquois dysplasia 1 (AR)	CANT1 (613165)	NM_001159773.2:c.277_278del (NP_001153245.1:p.Leu93fs)	Frameshift	17-78997344 CAG > C	0.01 (1/80)	1/0	6.6 × 10^−5^ (10/151,918)	0.006	0.003 (1/316)
rs797045079	208559 (2, LP)	Renal dysplasia (AR)	ACE (106180)	NM_000789.4:c.12_31del (NP_000780.1:p.Ser5fs)	Frameshift	17-63477105 CCTCGGGCCGCCGGGGGCCGG > C	0.01 (1/80)	1/0	2 × 10^−5^ (3/151,282)	0.002	0.003 (1/312)
rs777172978	1029383 (1, P)	Ullrich congenital muscular dystrophy 1 (AR)	COL6A2 (120240)	NM_058174.3:c.2554C > T (NP_478054.2:p.Gln852Ter)	Nonsense (stop gain)	21-46129288 C > T	0.01 (1/80)	1/0	7.2 × 10^−5^ (11/152,254)	0.006	0.006 (2/316)
rs121913572	14296 (2, P)	Merosin-deficient congenital muscular dystrophy (AR)	LAMA2 (156225)	NM_000426.4: c.7732C > T (NP_000417.3:p.Arg2578Ter)	Nonsense (stop gain)	6-129481422 C > T	0.01 (1/80)	1/0	4.6 × 10^−5^ (7/152,130)	0.004	0.003 (1/316)
rs121434505	8709 (1, P)	Fanconi anemia, complementation group E (AR)	FANCE (613976)	NM_021922.3: c.355C > T (NP_068741.1: p.Gln119Ter)	Nonsense (stop gain)	6-35455853 C > T	0.01 (1/80)	1/0	6.6 × 10^−5^ (1/152,228)	0.001	0.003 (1/316)
rs142637046	92361 (2, P)	Argininosuccinate lyase deficiency (AR)	ASL (608310)	NM_000048.4:c.446 + 1G > A	Splice donor	7-66083175 G > A	0.01 (1/80)	1/0	7.9 × 10^−5^ (12/151,984)	0.007	0.003 (1/316)
rs41286844	17038 (2, P)	Complement Component 6 deficiency (AR)	C8B (120960)	NM_000066.4:c.1282C > T (NP_000057.3:p.Arg428Ter)	Nonsense (stop gain)	1-56940965 G > A	0.03 (2/80)	2/0	0.001 (195/151,786)	0.005	0.02 (6/316)
rs1236367931	550946 (2, P)	Dysferlinopathy (AR)	DYSF (603009)	NM_007272.3:c.1471dup (NP_009203.2: p.Met491fs)	Frameshift	2-71535283 G > GA	0.08 (6/80)	6/0	1.3 × 10^−5^ (2/152,020)	4.9 × 10^−19^	0 (0/316)
rs7755898	12169 (2, P)	Classic congenital adrenal hyperplasia due to 21-hydroxylase deficiency (AR)	CYP21A2	NM_000500.9:c.955C > T (NP_000491.4:p.Gln319Ter)	Nonsense (stop gain)	6-32040421 C > T	0.08 (6/80)	6/0	0.004 (523/136,986)	7.6 × 10^−7^	0.004 (1/238)
rs202058123	430258 (1, LP)	Hereditary pancreatitis (AD)	CTRC (601405)	NM_007272.3:c.649G > A (NP_009203.2:p.Gly217Ser)	Missense	1-15445606 G > A	0.01 (1/80)	1/0	6.6 × 10^−5^ (10/152,212)	0.006	0.006 (2/316)
rs111033565	11876 (2, P)	Hereditary pancreatitis (AD)	PRSS1 (276000)	NM_002769.5:c.365G > A (NP_002760.1:p.Arg122His)	Missense	7-142751938 G > A	0.01 (1/80)	1/0	5.5 × 10^−5^ (8/145,178)	0.005	0 (0/302)

^a^ Number of stars by ClinVar and pathogenic level according to the ACMG-AMP guidelines (P-Pathogenic or LP-Likely Pathogenic). ^b^ Autosomal Dominant = AD; Autosomal Recessive = AR; X-linked Dominant = XLD; Multifactorial = M. ^c^ Chromosome-Position Reference > Alternative.

## Data Availability

The datasets generated and/or analyzed during the current study are available from the corresponding author upon reasonable request.

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
