# Peer review of "Comprehensive Genetic Analysis of Druze Provides Insights into Carrier Screening"

_genes, 2023, doi:10.3390/genes14040937_

Round 1
Reviewer 1 Report
The study performed by authors may be relevant for facilitating the genetic screening dedicated to small populations. Nevertheless, there are issues to be addressed:
· The experimental design should be better described: considering the screening purpose and the reported conclusion, was a power analysis performed before the selection of the sample size? This data could add robustness to the study cohorts and to the obtained results especially in light of improving the genetic screening. Moreover, the inclusion of 40 samples from the HGDP cohort deserves to be better explained (has it been included to compare Druze residing in Israel with other members of the same ethnic group?).
· Moreover, the materials and methods section should be improved. Although the methods have been already described in previous work, some information should be provided for the readers, concerning the employed sequencing technology, quality control metrics obtained in the present WES experiments and details concerning the bioinformatic pipelines. Authors should also specify why they relies only on Clinvar instead of including also the outcomes from other databases/tools (such as LOVD).
· Authors reported in the tables 1-2 the detected variants with the related rsID. Did the authors choose to focalize the study on known variants?
· The tables should also provide the n of subjects in which each variant has been identified and the related genotype.
Author Response
Reviewer # 1
Comment #1 - The experimental design should be better described: considering the screening purpose and the reported conclusion, was a power analysis performed before the selection of the sample size? This data could add robustness to the study cohorts and to the obtained results especially in light of improving the genetic screening. Moreover, the inclusion of 40 samples from the HGDP cohort deserves to be better explained (has it been included to compare Druze residing in Israel with other members of the same ethnic group?). The study was carried out on the maximal number of available cases who were eligible and consented to participate (in the WES study) and the data that are available from the HGDP dataset. Given the fact that we did not look for any association between the genetics and a phenotype or a trait and the limited number of Druze individuals world-wide, a priori power analysis seems not be of any consequence. The manner of recruitment of the Druze cohort in the HGDP dataset was expanded in the methods section.
Comment # 2- Moreover, the materials and methods section should be improved. Although the methods have been already described in previous work, some information should be provided for the readers, concerning the employed sequencing technology, quality control metrics obtained in the present WES experiments and details concerning the bioinformatic pipelines.- Expanded as requested. The original methods section included 254 words the current version – 606 and two supplementary tables.
Comment # 3- Authors should also specify why they relies only on Clinvar instead of including also the outcomes from other databases/tools (such as LOVD).
ClinVar is a relative broad and well establish database and it is part of the National Center for Biotechnology Information (NCBI). In addition, all variants in ClinVar are classified by the recommendation of the ACMG/AMP (https://www.sciencedirect.com/science/article/pii/S1098360021030318?via%3Dihub ). The ACMG/AMP suggested an advised and widely acceptable classification method (as mentioned by Reviewer #2). Therefore, ClinVar is a reliable, comprehensive, and consistent database, provides sufficient information for that type of studies. Moreover, we used gnomAD and dbSNP, for triple checking all the PVs.
Comment # 4- Authors reported in the tables 1-2 the detected variants with the related rsID. Did the authors choose to focalize the study on known variants? No. The variants that are reported are known variants (hence they have a rs number) but were not previously reported in the Druze population.
Comment # 5 - The tables should also provide the n of subjects in which each variant has been identified and the related genotype- The column Heterozygous/Homozygous was add to table 1 and table 2.
Reviewer 2 Report
Comprehensive genetic analysis of Druze provides insights into carrier screening
Avant et al. has performed genetic analysis of the Druze population and identified the disease causing variants based on the ClinVar pathogenic criteria. Further they have performed the statistical comparison with the non-Druze population. This study has high potential to extract out the variants that could be implemented for population carrier screening or prenatal screening for the Druze community.
However the authors have not performed the analysis properly. I would recommend the authors to make these changes that would significantly improve the manuscript:
Major:
-
Please provide the metadata of the population in the supplementary of WES as well as HGDP. Describing the age, sex, ethnicity etc.
-
Also provide a brief overview study population for readers to understand that in 118 Druze individuals. What is the sex ratio, average age, etc.
-
I would advise the authors to please detail the materials and methods section. Also add the supplementary data. Even after writing previously described need to briefly describe the different clans of Druze and what is the number would help the reader to understand about the methodology adopted by the authors.
-
Also when you write the whole exome sequencing as previously published protocols. Please provide a brief protocol of the WES as well as analysis.
-
I understand that ClinVar has a good source of variant filtering. But I advise the author to do the proper classification of the variants into pathogenic, likely pathogenic, benign, likely benign, and VUS based on American College of Medical Genetics and Genomics (ACMG-AMP) guidelines after filtering the variants. Off note these guidelines is not only for ACMG genes. It could be applied to all the Mendelian disorders.
-
Can you please explain more about what population has been taken from gnomAD?
-
How does the author use the dbSNP dataset? Since it is a resource of the SNP only, not the resource for population allele frequency. Please explain.
-
What population has been taken from dbSNP. I will advise the author to take the 1000 Genome Project population that include super and sub populations.
-
Why does rs565782662 does not have any variant frequency?
-
It is very surprising that in 2 variants that are autosomal dominant with very high frequency in general population and Druze community i.e. > 5% and in other > 10%. Why has it been marked as pathogenic? I would strongly advise author to follow ACMG guidelines to classify the variants otherwise these variants are not pathogenic in any way and might wrongly interpret it,
-
The last three variants have frequency > 50% which is wrongly calculated in the Druze AF column. Half of the population has the variant in the Druze community as well as the general population. It is very unlikely that it has any association with the disease even though it is multifactorial. Consider the scenariao where half of the world is suffering from Behcet. I would strongly recommend not interpreting the variant based on the ClinVar used the ACMG-AMP guidelines to classify these variants. They have 28 strict guidelines to classify any variant.
-
The Fisher exact p-value has been wrongly calculated. Please correct it.
-
I also advise the authors to make the bubble plot to show the allele frequency of the Pathogenic variant with the global population to compare and mark the variant that has significantly high frequency with the general populations calculated by the Fisher exact test.
Minor:
-
Line 22 “Druze individuals harbor recurring pathogenic variants (PV) in Autosomal recessive (AR) disorders”. Please expand the background in the Abstract such as why do they have AR disorder.
-
Line 22 write “Autosomal” in small letters as “autosomal”.
-
Line 24 no need of writing HGDP cohort again.
-
Line 25 Please change the line to this “Additionally we performed whole exome sequencing of 118 Druze individuals: 38 trios and 2 couples, representing geographically distinct clans”.
-
Line 39 please correct the reference [1)].
-
Line 42-44. This statement “These practices, com-42 bined with practicing marriages within extended families (47% of unions estimated as 43 consanguineous)” is correct but make it more scientific.
-
Line 48 “Druze include two PVs in ATM in Druze communities”. This is unclear ATM is the gene then please write it in italic as well define the disease ataxia telangiectasia.
-
Line 66 Please rewrite this statement “The study population were individuals who participated in the parent study”. Please explain a bit about the parent study.
-
Please check the citation 11.
-
At line 87 Instead of alpha level please write P-value.
-
In Table 1, the General Population AF column is written opposite instead of Allele count it is allele number and vice versa. Please correct it.
-
In Table 1 row 2, in the general population column what does Rs ID means here.
-
Same in Table 2, the General Population AF column is written opposite instead of Allele count it is allele number and vice versa. Please correct it.
-
Line 130, please write the full form of HMOs.
-
Please check the references.
Author Response
Major comments:
- Please provide the metadata of the population in the supplementary of WES as well as HGDP. Describing the age, sex, ethnicity etc. - Regarding the ethnicity, as is clearly stated, ALL cases are of Druze origin. Regarding age and sex- relevant data were added to the text.
- Also provide a brief overview study population for readers to understand that in 118 Druze individuals. What is the sex ratio, average age, etc. See response to comment # 1.
- I would advise the authors to please detail the materials and methods section. Also add the supplementary data. Even after writing previously described need to briefly describe the different clans of Druze and what is the number would help the reader to understand about the methodology adopted by the authors. – See response to comment # 2 to the previous reviewer.
- Also when you write the whole exome sequencing as previously published protocols. Please provide a brief protocol of the WES as well as analysis. - See response to comment # 2 to the previous reviewer.
- I understand that ClinVar has a good source of variant filtering. But I advise the author to do the proper classification of the variants into pathogenic, likely pathogenic, benign, likely benign, and VUS based on American College of Medical Genetics and Genomics (ACMG-AMP) guidelines after filtering the variants. Off note these guidelines is not only for ACMG genes. It could be applied to all the Mendelian disorders.- All variants in "ClinVar" are classified by the recommendation of the ACMG/AMP (https://www.sciencedirect.com/science/article/pii/S1098360021030318?via%3Dihub ) as written in ClinVar site (https://www.ncbi.nlm.nih.gov/clinvar/docs/clinsig/). Therefore, the classification of the variants in this study are as the reviewer advised.
- Can you please explain more about what population has been taken from gnomAD? - Thank you for raising this subject. We added the following clarification to the text: For each PV, general population allele count (AC) and general population allele number (AN) were retrieved from gnomAD (https://gnomad.broadinstitute.org/), as indicated by the total row in the Population Frequencies table. If AC and AN were missing, those values were extracted from dbSNP (https://www.ncbi.nlm.nih.gov/snp/), as indicated by the total column in the ALFA Allele Frequency table. Additionally, suitable AC and AN of the Middle East population were extracted from gnomAD, as indicated by the Middle East row in Population Frequencies table. Allele frequency (AF) was calculated by dividing AC by AN.
- How does the author use the dbSNP dataset? Since it is a resource of the SNP only, not the resource for population allele frequency. Please explain.- Please, see response to comment # 6. Additionally, dbSNP does present AF for the total population and for sub-populations. For example, please, check this link (https://www.ncbi.nlm.nih.gov/snp/rs147321712) under ALFA Allele Frequency section. Moreover, we only used dbSNP once- for rs312262845, since gnomAD's AC= 0 and the Fisher test yielded infinity results.
- What population has been taken from dbSNP. I will advise the author to take the 1000 Genome Project population that include super and sub populations.- We appreciate this suggestion. Unfortunately, before using dbSNP we did check the 1000 Genome Project population and there were no relevant data. Moreover, we only used dbSNP once- for rs312262845, since gnomAD's AC= 0 and the Fisher test yielded infinity results.
- Why does rs565782662 does not have any variant frequency? – We appreciate this comment. Indeed w,e did not find any variant frequency for this variant in both datasets- dbSNP and gnomAD. This is now clearly indicated in the table
- It is very surprising that in 2 variants that are autosomal dominant with very high frequency in general population and Druze community i.e. > 5% and in other > 10%. Why has it been marked as pathogenic? I would strongly advise author to follow ACMG guidelines to classify the variants otherwise these variants are not pathogenic in any way and might wrongly interpret it,- Please, see response to comment # 5 above.
- The last three variants have frequency > 50% which is wrongly calculated in the Druze AF column. Half of the population has the variant in the Druze community as well as the general population. It is very unlikely that it has any association with the disease even though it is multifactorial. Consider the scenariao where half of the world is suffering from Behcet. I would strongly recommend not interpreting the variant based on the ClinVar used the ACMG-AMP guidelines to classify these variants. They have 28 strict guidelines to classify any variant. – Thanks for this comment. In compliance with this comment any refence to these variants was deleted from the table
- The Fisher exact p-value has been wrongly calculated. Please correct it. – Done as suggested.
- I also advise the authors to make the bubble plot to show the allele frequency of the Pathogenic variant with the global population to compare and mark the variant that has significantly high frequency with the general populations calculated by the Fisher exact test. – Thank you for this suggestion. Unlike the reviewer we feel that adding such an additional plot would not add any information that would serve to better clarify the main message of this study. Therefore, we would like to avoid this addition.
Minor:
- Line 22 “Druze individuals harbor recurring pathogenic variants (PV) in Autosomal recessive (AR) disorders”. Please expand the background in the Abstract such as why do they have AR disorder. - Done
- Line 22 write “Autosomal” in small letters as “autosomal”. - Done
- Line 24 no need of writing HGDP cohort again.- We use the term HGDP cohort when we refer to the analyzed cohort. This is not a stand-alone word in the abstract. To highlight this fact, we added a hyphen between the two words
- Line 25 Please change the line to this “Additionally we performed whole exome sequencing of 118 Druze individuals: 38 trios and 2 couples, representing geographically distinct clans”.- Done
- Line 39 please correct the reference [1)]. – Done
- Line 42-44. This statement “These practices, combined with practicing marriages within extended families (47% of unions estimated as 43 consanguineous)” is correct but make it more scientific.- Done
- Line 48 “Druze include two PVs in ATM in Druze communities”. This is unclear ATM is the gene then please write it in italic as well define the disease ataxia telangiectasia.- Done
- Line 66 Please rewrite this statement “The study population were individuals who participated in the parent study”. Please explain a bit about the parent study. - Done
- Please check the citation 11. This is the original citation and it is adequately represented in the reference list. Strauss KA, Gonzaga-Jauregui C, Brigatti KW, Williams KB, King AK, Van Hout C, Robinson DL, Young M, Praveen K, Heaps AD, Kuebler M, Baras A, Reid JG, Overton JD, Dewey FE, Jinks RN, Finnegan I, Mellis SJ, Shuldiner AR, Puffenberger EG. Genomic diagnostics within a medically underserved population: efficacy and implications. Genet Med. 2018 Jan;20(1):31-41. doi: 10.1038/gim.2017.76. Epub 2017 Jul 20. PMID: 28726809. In this publication it is elaborated on the WES technology used at Regenron.
- At line 87 Instead of alpha level please write P-value. – Done
- In Table 1, the General Population AF column is written opposite instead of Allele count it is allele number and vice versa. Please correct it. -Done
- In Table 1 row 2, in the general population column what does Rs ID means here.- It simply means rs number. In compliance with this comment, Rs ID was changed to Rs Number in both tables.
- Same in Table 2, the General Population AF column is written opposite instead of Allele count it is allele number and vice versa. Please correct it. - Done
- Line 130, please write the full form of HMOs. – Done
- Please check the references. – Done
Round 2
Reviewer 2 Report
1. I still recommend author to classify their variant based ACMG-AMP classification. Even author explained that ClinVar has followed ACMG and AMP guidelines. I would suggest to please do it again to get the consensus. Because different submitter has annotated same variant as benign as well pathogenic. So using ACMG-AMP author can be sure whether it is pathogenic or benign variant or VUS. I searched for one variant rs5030737, whose global allele frequency is 5%. It is clearly polymorphic, large number of healthy individuals has this variant in global population in homozygote state. I donot think it is disease causing variant.
2. I suggested to make bubble plot not for the author but for the reader who could just go through the figure and can understand rather than reading the whole table. It is always better to understand the data graphically rather than going through the numbers.
Author Response
Reviewer # 1
Comment #1 - I still recommend author to classify their variant based ACMG-AMP classification. Even author explained that ClinVar has followed ACMG and AMP guidelines. I would suggest to please do it again to get the consensus. Because different submitter has annotated same variant as benign as well pathogenic. So using ACMG-AMP author can be sure whether it is pathogenic or benign variant or VUS. I searched for one variant rs5030737, whose global allele frequency is 5%. It is clearly polymorphic, large number of healthy individuals has this variant in global population in homozygote state. I donot think it is disease causing variant.- We re-evaluated all variants by the ACMG – AMP guidelines, independent of ClinVar. As a result, one of the variants was deleted from the table. These additional analyses are clearly shown in the methods and results section.
Comment # 2- I suggested to make bubble plot not for the author but for the reader who could just go through the figure and can understand rather than reading the whole table. It is always better to understand the data graphically rather than going through the numbers.- Two Bubble plots were created and inserted into the text where appropriate, as suggested.
Round 3
Reviewer 2 Report
The authors has successfully addressed the comments.